# Effect of postmastectomy radiotherapy on T1-2N1M0 triple-negative breast cancer

**Lin-Yu Xia** [1] *, **Wei-Yun Xu** [2], **Yan Zhao** [3]

**1** Department of Thyroid and Breast Surgery, The First Affiliated Hospital of Chengdu Medical College, Chengdu, Sichuan, China, **2** Department of Breast Surgery, Mianyang Central Hospital, Mianyang, Sichuan, China, **3** Department of Breast Surgery, Sichuan Mianyang 404 Hospital, Mianyang, Sichuan, China

* lylc1023@163.com

**Data Availability Statement:** All data are available from the Surveillance, Epidemiology, and End Results (SEER) Program (www.seer.cancer.gov) SEER*Stat Database: Incidence - SEER 18 Regs Custom Data (with additional treatment fields), Available at:https://seer.cancer.gov/data/.

## Abstract

### Background

The effect of postmastectomy radiotherapy (PMRT) on T1-2N1M0 triple-negative breast cancers (TNBC) remains unclear. The population-based study aimed to investigate the survival outcomes of T1-2N1M0 TNBC patients who underwent PMRT or not.

### Methods

We selected 1743 patients with T1-2N1M0 TNBC who underwent mastectomy between 2010 and 2015 through the Surveillance, Epidemiology and End Results (SEER) database. After propensity score matching (PSM), the PMRT and no-PMRT groups consisted of 586 matched patients, respectively. The Kaplan-Meier method was applied to calculate breast cancer-specific survival (BCSS) and cox proportional hazard model was used to determine the prognostic factors of T1-2N1M0 TNBC.

### Results

The 5-year BCSS for the PMRT and no-PMRT groups was 79.1% and 74.7%, respectively. Analysis showed that in patients with three nodes positive, radiotherapy could significantly improve BCSS (HR = 0.396, 95% CI = 0.175–0.900, $P$ = 0.027), but it brought no significant advantage in BCSS in patients with one or two nodes positive (HR = 1.061, 95% CI = 0.725–1.552, $P$ = 0.761; HR = 0.657, 95% CI = 0.405–1.065, $P$ = 0.088). In addition, PMRT improves the BCSS in TNBC patients with T2 tumor concomitant with three positive lymph nodes (HR = 0.343, 95% CI = 0.132–0.890, $P$ = 0.028).

### Conclusion

TNBC patients with T2 tumor concomitant with three positive lymph nodes can benefit from PMRT.

**Funding:** The author(s) received no specific funding for this work.

**Competing interests:** The authors have declared that no competing interests exist.

## Introduction

TNBC is defined as breast cancer that lacks estrogen receptor, progesterone receptor expression and human epidermal growth factor receptor 2 (HER-2) over-expression or gene amplification, accounting for 10% ~ 15% of all breast cancer [1–3]. As a special type of breast cancer, TNBC has a special biological behavior, with a high degree of malignancy and early recurrence and metastasis [4–6].

Due to the lack of therapeutic targets, TNBC does not benefit from endocrine therapy and anti-HER-2 targeted therapy. At present, the main clinical treatment is surgery combined with radiotherapy and chemotherapy. Chemotherapy is one of the main means of systemic treatment for TNBC patients, while surgery and radiotherapy are the main local treatment for TNBC patients. Currently, the indications for adjuvant radiotherapy after mastectomy are primary tumor diameter≥ 5cm or the number of axillary lymph node metastases ≥ 4, while there is still controversy about whether adjuvant radiotherapy is necessary for T1-2 breast cancer patients with 1 to 3 axillary lymph node metastasis [7–9]. Studies have found no significant improvement in DFS and OS in T1-2N1M0 TNBC patients receiving PMRT [10, 11]. On the contrary, other studies have confirmed that T1-2N1M0 TNBC patients can benefit from PMRT [12, 13]. In the available evidence, the effect of PMRT on T1-2N1M0 TNBC patients is contradictory, and the value of PMRT on T1-2N1M0 TNBC patients needs to be further clarified. In this study, we investigated the effect of PMRT on BCSS In T1-2N1M0 TNBC patients and performed subgroup analyses to determine which patients could benefit from PMRT.

## Materials and methods

### Patients

We collected data from the SEER database for this study. The inclusion criteria included: (1) female; (2) 20–79 years old; (3) diagnosed with TNBC from 2010 to 2015; (4) T1-2N1M0; (5) The mastectomy was performed. Exclusion criteria included:(1) patients who did not undergo axillary dissection; (2) patients with unknown clinicopathologic characteristics; (3) patients without radiotherapy records or chemotherapy records.

We collected the following variables: age, year of diagnosis, race, marital status, histology, histological grade, number of lymph nodes (LNs) positive, chemotherapy record, radiotherapy record, follow-up time, and vital status.

### Statistical analysis

The main outcome of interest was BCSS, which was calculated from the date of diagnosis to the date of death due to breast cancer. The clinicopathologic characteristics of the PMRT and no-PMRT groups were compared through the $X^2$ test. One-to-one (1:1) PSM was used to create a matched dataset to balance the baseline characteristics between the two groups. The survival curve of BCSS was plotted by the Kaplan-Meier product-limit method and compared by the log-rank test. A Cox proportional hazards regression model was used to analyze the prognostic factors associated with BCSS. P values were two-sided, and $P < 0.05$ was considered statistically significant. These analyses were performed using the SPSS version 20.0 software package (IBM SPSS Statistics, Chicago, IL, US) and the R Project (R version 3.6.2 for Windows).

### Ethics statement

The study obtained data from the SEER database and did not require ethical consent because all data were fully anonymized and were publicly available.

## Results

### Patient characteristics

1743 patients diagnosed with T1-2N1M0 TNBC from 2010 to 2015 were recruited for the study. They were divided into the PMRT (789, 45.27%) and no-PMRT (954, 54.73%) groups. Compared with the no-PMRT group, patients in the PMRT group were older, had higher grades, had larger tumors, had more positive lymph nodes, were more likely to receive chemotherapy, and had a higher proportion of invasive ductal carcinoma (P<0.05). After PSM, the PMRT and no-PMRT groups consisted of 586 matched patients, respectively. There were no significant differences between the variables of the two groups after PSM (*P*>0.05). Table 1 summarizes the patient characteristics of the two groups.

### Prognostic factors associated with BCSS

We compared the BCSS of the PMRT and no-PMRT groups. With a median follow-up of 69.5 months, the 5-year BCSS for the PMRT and no-PMRT groups was 79.1% and 74.7%, respectively (Log-rank *P* = 0.166, Fig 1A). We studied the prognostic factors associated with BCSS. Univariate analysis indicated that BCSS was related to marital status, tumor size, and the number of positive LNs (all *P* < 0.05). Multivariate analysis showed that tumor size (*P* = 0.007) and two positive LNs (*P* = 0.005) were associated with BCSS. However, chemotherapy and radiotherapy had no statistically significant impact on the BCSS of T1-2N1M0 TNBC (all *P* > 0.05) (Table 2).

**Table 1. Baseline characteristics of the study population and tumor.**

| Characteristics | | before PSM [a] | | | after PSM | | |
|---|---|---|---|---|---|---|---|
| | | PMRT [b] (n,%) | No-PMRT (n,%) | *P* | PMRT (n,%) | No-PMRT (n,%) | *P* |
| **No. of patients** | | 789 | 954 | | 586 | 586 | |
| **Age (years)** | 20–49 | 375(47.53) | 324(33.96) | <0.001 | 242(41.30) | 243(41.47) | 0.953 |
| | 50–80 | 414(52.47) | 630(66.04) | | 344(58.70) | 343(58.53) | |
| **Race** | White | 547(69.33) | 713(74.74) | 0.042 | 450(76.79) | 444(75.77) | 0.588 |
| | Black | 162(20.53) | 159(16.67) | | 99(16.89) | 96(16.38) | |
| | Other | 80(10.14) | 82(8.59) | | 37(6.32) | 46(7.85) | |
| **Marital status** | Married | 477(60.46) | 583(61.11) | 0.78 | 366(62.46) | 362(61.77) | 0.810 |
| | Not married | 312(39.54) | 371(38.89) | | 220(37.54) | 224(38.23) | |
| **Histology** | IDC | 72(91.25) | 834(87.42) | 0.010 | 535(91.30) | 533(90.96) | 0.837 |
| | others | 69(8.75) | 120(12.58) | | 51(8.70) | 53(9.04) | |
| **Grade** | I+II | 96(12.17) | 165(17.30) | 0.001 | 70(11.95) | 67(11.43) | 0.785 |
| | III+IV | 693(87.83) | 739(82.70) | | 516(88.05) | 519(88.57) | |
| **Tumor stage** | T1 | 227(28.77) | 363(38.05) | <0.001 | 174(29.69) | 172(29.35) | 0.898 |
| | T2 | 562(71.23) | 591(61.95) | | 412(70.31) | 414(70.65) | |
| **No. of LNs [c] positive** | 1 | 434(55.01) | 625(65.51) | <0.001 | 362(61.77) | 376(64.16) | 0.693 |
| | 2 | 209(26.49) | 248(26.00) | | 157(26.79) | 146(24.91) | |
| | 3 | 146(18.50) | 81(8.49) | | 67(11.43) | 64(10.93) | |
| **Chemotherapy** | yes | 767(97.21) | 776(81.34) | <0.001 | 565(96.42) | 566(96.59) | 0.874 |
| | no | 22(2.79) | 178(18.66) | | 21(3.58) | 20(3.41) | |

[a] PSM = propensity score matching.

[b] PMRT = postmastectomy radiotherapy.

[c] LN = lymph node.

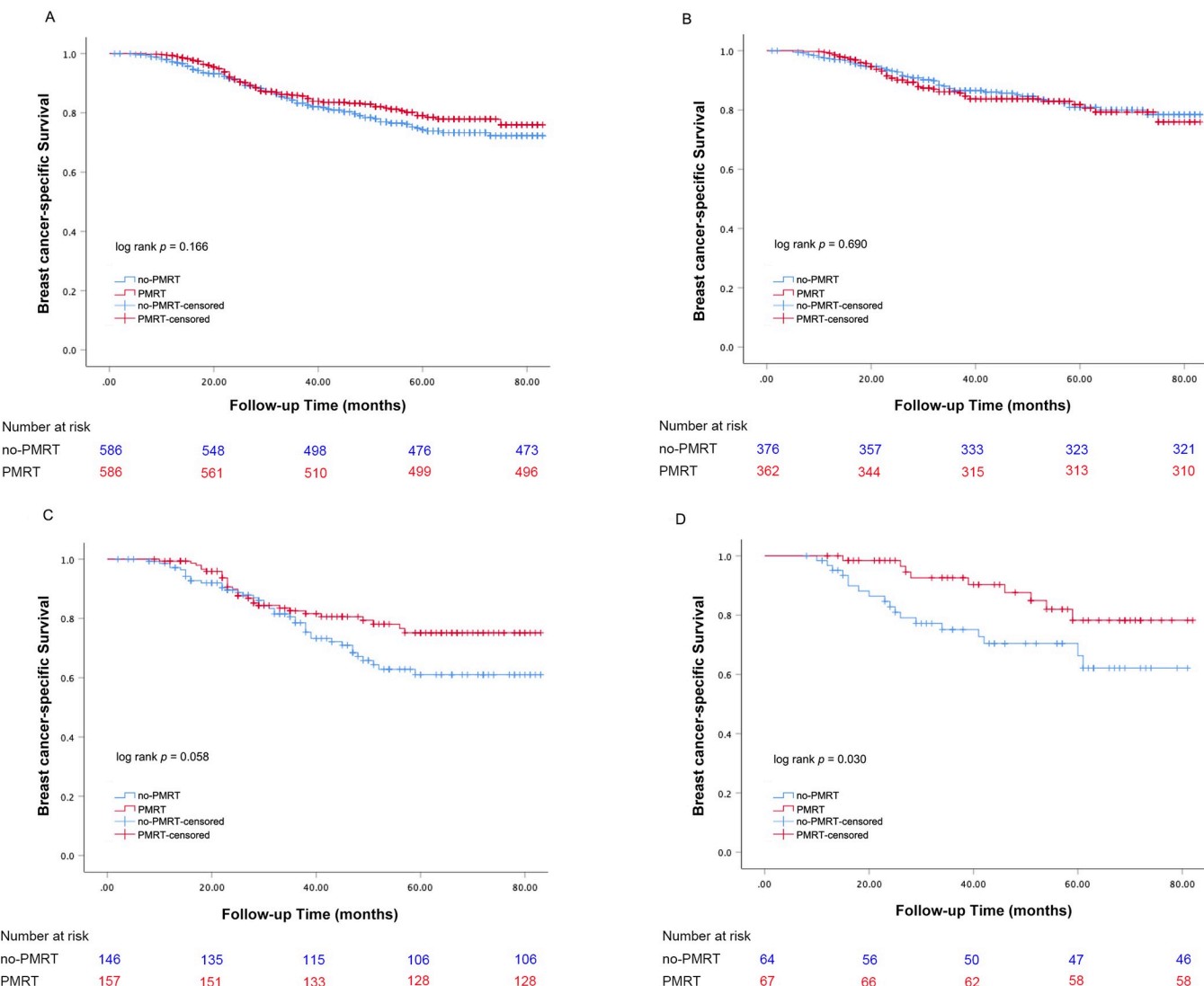

**Fig 1.** Kaplan-Meier curves of BCSS for T1-2N1M0 TNBC patients with and without PMRT: (A) all patients; (B) patients with one positive LN; (C) patients with two positive LNs; (D) patients with three positive LNs.

## Subgroup analyses of BCSS

We conducted subgroup analyses to determine the effect of radiotherapy on BCSS for T1-2N1M0 TNBC patients in different features (Table 3; Fig 2). Univariate analysis indicated that PMRT could improve the BCSS in patients with three nodes positive (HR = 0.423, 95% CI = 0.190–0.941, $P$ = 0.035), but it brought no significant advantage in BCSS in other patients. Multivariate analysis also showed that PMRT could improve the BCSS in patients with three nodes positive (HR = 0.396, 95% CI = 0.175–0.900, $P$ = 0.027; Fig 1D), but could not improve BCSS in patients with one or two nodes positive (HR = 1.061, 95% CI = 0.725–1.552, $P$ = 0.761; HR = 0.657, 95% CI = 0.405–1.065, $P$ = 0.088) (Fig 1B and 1C).

Furthermore, to investigate the effect of PMRT on BCSS for TNBC patients with certain tumor sizes and the number of positive LNs, we crossed tumor size with the number of positive LNs to divide patients into six subgroups (S1 Fig). Multivariate analysis indicated that PMRT could improve the BCSS in TNBC patients with T2 tumor concomitant with three

**Table 2. Prognostic factors for BCSS in univariate and multivariate analysis.**

| Characteristics | | Univariate | | Multivariate | |
|---|---|---|---|---|---|
| | | HR(95%CI) | *P* | HR(95%CI) | *P* |
| **Age (years)** | 20–49 | Ref. | Ref. | Ref. | Ref. |
| | 50–80 | 1.241(0.933–1.650) | 0.138 | 1.244(0.933–1.660) | 0.137 |
| **Race** | White | Ref. | Ref. | Ref. | Ref. |
| | Black | 1.013(0.699–1.467) | 0.947 | 0.932(0.639–1.358) | 0.713 |
| | Other | 0.708(0.374–1.341) | 0.289 | 0.686(0.361–1.305) | 0.251 |
| **Marital status** | Not Married | Ref. | Ref. | Ref. | Ref. |
| | Married | 0.739(0.560–0.975) | 0.033 | 0.766(0.575–1.019) | 0.067 |
| **Histology** | IDC | Ref. | Ref. | Ref. | Ref. |
| | others | 0.560(0.305–1.028) | 0.061 | 0.583(0.316–1.075) | 0.084 |
| **Grade** | I+II | Ref. | Ref. | Ref. | Ref. |
| | III+IV | 1.145(0.722–1.817) | 0.564 | 1.119(0.701–1.784) | 0.638 |
| **Tumor stage** | T1 | Ref. | Ref. | Ref. | Ref. |
| | T2 | 1.631(1.168–2.277) | 0.004 | 1.589(1.135–2.224) | 0.007 |
| **No. of LNs [a] positive** | 1 | Ref. | Ref. | Ref. | Ref. |
| | 2 | 1.590(1.175–2.152) | 0.003 | 1.543(1.138–2.093) | 0.005 |
| | 3 | 1.401(0.919–2.138) | 0.117 | 1.275(0.833–1.953) | 0.263 |
| **Chemotherapy** | no | Ref. | Ref. | Ref. | Ref. |
| | yes | 1.350(0.599–3.041) | 0.469 | 1.344(0.592–3.050) | 0.480 |
| **Radiotherapy** | no | Ref. | Ref. | Ref. | Ref. |
| | yes | 0.823(0.624–1.086) | 0.168 | 0.800(0.605–1.056) | 0.115 |

[a] LN = lymph node.

positive LNs (HR = 0.343, 95% CI = 0.132–0.890, *P* = 0.028). However, other subgroups could not benefit from PMRT (Fig 3).

## Discussion

We used the SEER database to study the influence of PMRT on BCSS in T1-2N1M0 TNBC patients and conducted a subgroup analysis to find out the subgroups that could benefit from PMRT. The study showed that T1-2N1M0 TNBC patients with three positive nodes could get BCSS improvement from radiotherapy, in which the patients with T2 tumor concomitant with three positive LNs benefit significantly. Our study provided evidence for the management of PMRT for T1-2N1M0 TNBC patients.

Currently, the management of PMRT in T1-2N1M0 breast cancer patients remains controversial. Some studies have confirmed that such patients can benefit from PMRT [14–16], while some studies have drawn the opposite conclusion [11, 17, 18]. Based on the current evidence, the recommendations of PMRT for T1-2N1M0 breast cancer were obviously different [7–9]. Given the high invasiveness and early recurrence of TNBC, local radiotherapy is particularly important. A study explored the effect of PMRT on T1-2N1M0 patients according to molecular typing, and the results showed that PMRT could reduce the local recurrence rate of TNBC patients [19]. Gabos et al. also confirmed that PMRT could reduce the local recurrence rate, especially for women with the T1-2N0 TNBC subtype [20]. However, as with other subtypes of breast cancer, current recommendations for PMRT in T1-2N1M0 TNBC patients are controversial.

**Table 3. Subgroup analysis of BCSS in univariate and multivariate analysis.**

| Characteristics | | Univariate | | Multivariate | |
|---|---|---|---|---|---|
| | | HR(95%CI) | P | HR(95%CI) | P |
| **Age (years)** | 20–49 | 1.010(0.642–1.589) | 0.996 | 0.960(0.609–1.514) | 0.861 |
| | 50–80 | 0.724(0.510–1.029) | 0.071 | 0.725(0.509–1.031) | 0.074 |
| **Race** | White | 0.776(0.567–1.062) | 0.113 | 0.746(0.544–1.022) | 0.068 |
| | Black | 1.152(0.587–2.260) | 0.680 | 1.142(0.581–2.246) | 0.700 |
| | Other | 0.604(0.154–2.363) | 0.469 | 0.844(0.186–3.833) | 0.826 |
| **Marital status** | Not Married | 0.825(0.542–1.257) | 0.371 | 0.835(0.547–1.275) | 0.404 |
| | Married | 0.822(0.569–1.188) | 0.297 | 0.781(0.539–1.131) | 0.191 |
| **Histology** | IDC | 0.814(0.612–1.083) | 0.157 | 0.793(0.596–1.056) | 0.112 |
| | others | 0.954(0.289–3.147) | 0.938 | 0.963(0.264–3.508) | 0.954 |
| **Grade** | I+II | 0.647(0.264–1.583) | 0.340 | 0.618(0.248–1.541) | 0.302 |
| | III+IV | 0.844(0.631–1.129) | 0.254 | 0.819(0.611–1.097) | 0.180 |
| **Tumor stage** | T1 | 1.019(0.564–1.841) | 0.950 | 1.039(0.574–1.881) | 0.900 |
| | T2 | 0.775(0.566–1.062) | 0.112 | 0.753(0.549–1.033) | 0.078 |
| **No. Of LNs [a] positive** | 1 | 1.080(0.739–1.580) | 0.691 | 1.061(0.725–1.552) | 0.761 |
| | 2 | 0.633(0.392–1.022) | 0.061 | 0.657(0.405–1.065) | 0.088 |
| | 3 | 0.423(0.190–0.941) | 0.035 | 0.396(0.175–0.900) | 0.027 |
| **Chemotherapy** | no | 0.163(0.019–1.402) | 0.098 | 0.061(0.004–0.938) | 0.055 |
| | yes | 0.866(0.654–1.146) | 0.314 | 1.191(0.899–1.578) | 0.224 |

[a] LN = lymph node.

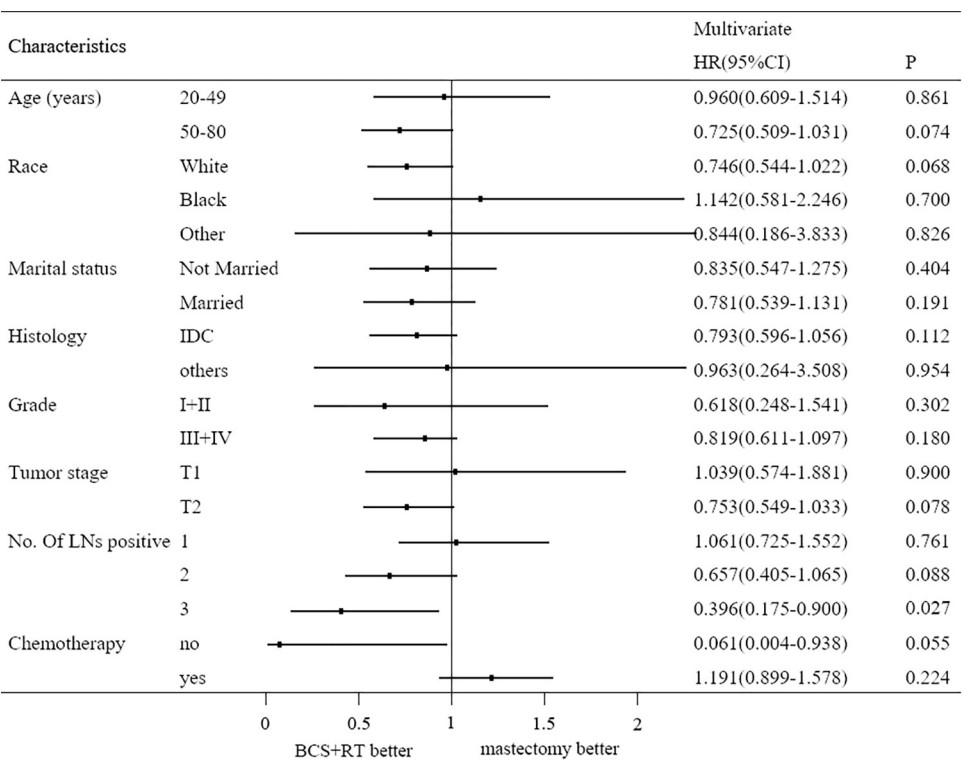

**Fig 2. The forest plot of HR for BCSS in PMRT vs. no-PMRT group in subgroup analysis.**

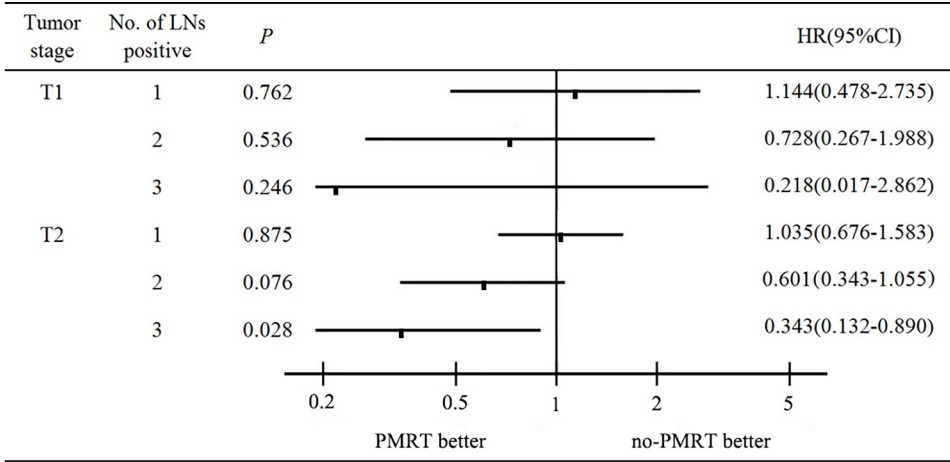

| Tumor stage | No. of LNs positive | P | | HR(95%CI) |
|---|---|---|---|---|
| T1 | 1 | 0.762 | | 1.144(0.478-2.735) |
| | 2 | 0.536 | | 0.728(0.267-1.988) |
| | 3 | 0.246 | | 0.218(0.017-2.862) |
| T2 | 1 | 0.875 | | 1.035(0.676-1.583) |
| | 2 | 0.076 | | 0.601(0.343-1.055) |
| | 3 | 0.028 | | 0.343(0.132-0.890) |

**Fig 3. The forest plot of HR for BCSS in PMRT vs. no-PMRT group in T1-2N1M0 TNBC patients stratified by tumor size and the number of positive lymph nodes.**

As previously reported [12, 13], TNBC had a low incidence of LN involvement. In our study, 62.97% of patients had one positive lymph node, and only 11.18% of patients had three positive lymph nodes. The number of patients with 3 nodes is fewer than 2 and 1 nodes. The small numbers may be the reason for the hazard ratio of 2 positive nodes larger than 3 nodes (Table 2). Tumor size and the number of axillary lymph node metastases have been proved to be closely related to the recurrence and prognosis of breast cancer [21, 22]. In our study, tumor size and the number of positive LNs were associated with BCSS. The effect of PMRT mainly depends on the comprehensive consideration of tumor size and the number of positive LNs [23]. Secondly, it is related to other high-risk factors such as young age and positive vascular tumor thrombus. Accordingly, we not only divided the subgroup analysis according to the clinicopathological characteristics but also performed subgroup analysis by crossing tumor size with the number of positive LNs, which proved that only BCSS in T2 patients with three positive LNs could benefit from PMRT. Zhang et al. studied the effect of PMRT on the survival of T1-4N1-N3M0 patients, and the results showed that there was no difference in BCSS between PMRT and non-PMRT cohort in the T1-2N1 subgroup (*P* = 0.191) [24], which was consistent with our results, but they did not conduct further stratified analysis. Another study included 675 T1-2N1M0 TNBC patients and subgroup analysis was performed based on the number of positive lymph nodes. There were 312 patients in the PMRT group and 363 patients in the no-PMRT group, after a median follow-up of 37 months, PMRT was independently associated with increased OS, but there were no significant differences in OS or BCSS between the groups stratified by the number of positive lymph nodes [25]. The reason why they are inconsistent with our conclusion may be that they included fewer patients and had a shorter follow-up time. In addition, there was a significant difference between the two groups in their study. Patients in the PMRT group had a heavier nodal burden, and the proportion of chemotherapy was higher than that in the no-PMRT group. There was no significant difference in baseline characteristics between the two groups in our study. This may also be the reason for our inconsistent conclusions.

Radiotherapy can not only eliminate the residual lesions and reduce the recurrence rate of the disease but also bring a series of complications, such as arm edema, cardiopulmonary radiation damage, pneumonia, and so on [26–28]. We need to comprehensively consider the benefits and side effects of radiotherapy for patients, identify the subgroups that can really benefit

from radiotherapy, and optimize personalized treatment strategies. Our study conducted subgroup analyses based on clinicopathological characteristics and proves that radiotherapy did not provide survival benefits for T1-2 TNBC patients with 1 or 2 positive LNs. Chen et al. also showed that T1-2 breast cancer patients with one or two positive LNs could not benefit from PMRT [29]. It may be that the tumor load and recurrence risk of these patients is low. Radiotherapy has limited significance to improve their survival and will instead bring side effects such as arm edema and cardiopulmonary radiation damage. For these patients, we should carefully choose radiotherapy to avoid overtreatment.

Our study has some limitations. First, we did not study the recurrence rate of patients because the SEER database lacks recurrence data. Second, HER-2 status in the SEER database was only available since 2010, so our follow-up period was relatively short. Third, the SEER database does not provide detailed radiotherapy protocols.

## Conclusion

In a word, our findings suggested that PMRT could significantly improve BCSS in T1-2N1M0 TNBC patients with three nodes positive, but it brought no significant advantage in BCSS in patients with one or two nodes positive. Patients with T2 tumor concomitant with three positive LNs benefit significantly from radiotherapy and should be advised to receive radiotherapy. Patients with one positive lymph node should not receive radiotherapy. For other patients, radiotherapy should be chosen carefully in combination with other high-risk factors.

## Supporting information

**S1 Fig.** Kaplan-Meier curves of BCSS for TNBC patients with and without PMRT:(A) patients with T1 tumor and one positive LN; (B) patients with T1 tumor and two positive LNs; (C) patients with T1 tumor and three positive LNs; (D) patients with T2 tumor and one positive LN; (E) patients with T2 tumor and two positive LNs; (F) patients with T2 tumor and three positive LNs.
(TIF)

## Author Contributions

**Data curation:** Lin-Yu Xia.

**Formal analysis:** Lin-Yu Xia.

**Methodology:** Lin-Yu Xia, Wei-Yun Xu.

**Writing – original draft:** Lin-Yu Xia, Yan Zhao.

**Writing – review & editing:** Lin-Yu Xia, Wei-Yun Xu, Yan Zhao.

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
