## [Decision Letter · Decision Letter 0]

12 May 2022

PONE-D-22-00975Effect of postmastectomy radiotherapy on T1-2N1M0 triple-negative breast cancer.PLOS ONE

Dear Dr. Xia,

Thank you for submitting your manuscript to PLOS ONE. After careful consideration, we feel that it has merit but does not fully meet PLOS ONE’s publication criteria as it currently stands. Therefore, we invite you to submit a revised version of the manuscript that addresses the points raised during the review process.

We look forward to receiving your revised manuscript.

Kind regards,

Sudeep Gupta, M.D.

Academic Editor

PLOS ONE

Journal Requirements:

Reviewers' comments:

Reviewer's Responses to Questions

**Comments to the Author**

1. Is the manuscript technically sound, and do the data support the conclusions?

Reviewer #1: Yes

2. Has the statistical analysis been performed appropriately and rigorously? 

Reviewer #1: Yes

3. Have the authors made all data underlying the findings in their manuscript fully available?

Reviewer #1: Yes

4. Is the manuscript presented in an intelligible fashion and written in standard English?

Reviewer #1: Yes

5. Review Comments to the Author

Reviewer #1: The authors try to understand the benefit of adding adjuvant radiation after mastectomy in patients with 1-3 nodes positive in TNBC patients. The authors present a sound hypothesis, methodology and statistical analysis. The manuscript has an intuitive flow. There are however, some issues that the authors need to address to bring out more clarity to their findings.

1. It is counterintuitive that the number of patients with 3 nodes are fewer than 2 and 1 nodes. TNBC patients usually present with a heavier nodal burden. Can the authors please explain this in their discussion section? The small numbers may be the reason for the hazard ratio of 2 positive nodes larger than 3 nodes (table 2)

2. Is it possible for the authors to also inform the readers on the number of dissected nodes? In some studies that have shown a benefit of RT in a similar cohort, the benefit has been attributed to inadequate surgery (fewer dissected nodes).

3. The Survival curves for all the figures would benefit from addition of numbers at risk at the bottom to give the readers a better perspective of the outcomes

4. Line 157-161: The authors have quoted Zhang et al. here who have shown a divergent result compared to the current study. The reason given is smaller patient numbers. The numbers in the study by Zhang et al. are more than the current study. So can the authors think of a better explanation or provide more clarity on their statement?

6. PLOS authors have the option to publish the peer review history of their article (what does this mean?). If published, this will include your full peer review and any attached files.

Reviewer #1: **Yes: **Dr. Rima Sanjay Pathak

---

## [Author Response · Author response to Decision Letter 0]

7 Jun 2022

Dear Reviewer,

 Thank you for your kind comments concerning our manuscript entitled “ Effect of postmastectomy radiotherapy on T1-2N1M0 triple-negative breast cancer. (PONE-D-22-00975)”. Those comments are all valuable and very helpful for revising and improving our paper. We revised the manuscript in accordance with your comments carefully. Revised portions are marked in yellow on the paper.

Here below is our description of the revision according to your comments.

Reviewer 1:

1. Is the manuscript technically sound, and do the data support the conclusions?

Reviewer #1: Yes

Response: Thank you for your careful review of our manuscript. 

2. Has the statistical analysis been performed appropriately and rigorously?

Reviewer #1:  Yes

Response: Thank you for your comment. 

3. Have the authors made all data underlying the findings in their manuscript fully available?

Reviewer #1: Yes

Response: Thanks for your affirmation of our manuscript. 

4.Is the manuscript presented in an intelligible fashion and written in standard English?

Reviewer #1: Yes

Response: Thanks for your affirmation of our manuscript. Our manuscript has been edited by a native English speaking editor, and we have also carefully proofread it to minimize typographical and grammatical errors.

5.The authors try to understand the benefit of adding adjuvant radiation after mastectomy in patients with 1-3 nodes positive in TNBC patients. The authors present a sound hypothesis, methodology and statistical analysis. The manuscript has an intuitive flow. There are however, some issues that the authors need to address to bring out more clarity to their findings.It is counterintuitive that the number of patients with 3 nodes are fewer than 2 and 1 nodes. TNBC patients usually present with a heavier nodal burden. Can the authors please explain this in their discussion section? The small numbers may be the reason for the hazard ratio of 2 positive nodes larger than 3 nodes (table 2).

Response: Thank you very much for your question. We checked the data carefully and got the same result. We also referenced other literature and found that our results were consistent with those of other studies. Wang et al. [1] also collected data from SEER. In their study, there were 887 TNBC patients, of which 54.79% were positive for one lymph node, 27.96% were positive for two lymph nodes, and 17.25% were positive for three lymph nodes. Bassam [2] and Chen et al. [3] also confirmed that TNBC patients had a low incidence of LN involvement. We have explained this in the discussion section.

[1] Wang XY, Xu YY, Guo SS, Zhang JX, Abe M, Tan HS, et al. T1-2N1M0 triple-negative breast cancer patients from the SEER database showed potential benefit from postmastectomy radiotherapy. Oncol Lett. 2020;19(1): 735-744. doi:10.3892/ol.2019.11139.

[2] Bassam S. Abdulkarim, Julie Cuartero, John Hanson, Jean Deschênes, David Lesniak, and Siham Sabri. Increased Risk of Locoregional Recurrence for Women With T1-2N0 Triple-Negative Breast Cancer Treated With Modified Radical Mastectomy Without Adjuvant Radiation Therapy Compared With Breast-Conserving Therapy.J Clin Oncol. 2011 Jul 20; 29(21): 2852–2858.doi: 10.1200/JCO.2010.33.4714

[3] Chen X, Yu X, Chen J, Yang Z, Shao Z, Zhang Z, et al. Radiotherapy can improve the disease-free survival rate in triple-negative breast cancer patients with T1-T2 disease and one to three positive lymph nodes after mastectomy. Oncologist. 2013;18:141-7. doi: 10.1634/theoncologist.2012-0233. 

6. Is it possible for the authors to also inform the readers on the number of dissected nodes? In some studies that have shown a benefit of RT in a similar cohort, the benefit has been attributed to inadequate surgery (fewer dissected nodes).

Response: It's a pity that we didn't collect this data at the initial stage of design, which is indeed a limitation of our research, but we think what you said is very reasonable, so we hope to continue to carry out this research in the future to make our research more perfect.

7. The Survival curves for all the figures would benefit from addition of numbers at risk at the bottom to give the readers a better perspective of the outcomes

 Response: This is a very good suggestion to make the image more intuitive. We have modified it according to your suggestion.

8. Line 157-161: The authors have quoted Zhang et al. here who have shown a divergent result compared to the current study. The reason given is smaller patient numbers. The numbers in the study by Zhang et al. are more than the current study. So can the authors think of a better explanation or provide more clarity on their statement?

Response: Thanks for your question. We quoted two studies here, the first by Zhang et al. [1] including 4398 TNBC patients, and their conclusions are consistent with our findings. The second study by Zhang et al. [2] included 675 T1-2N1M0 TNBC patients and there were 312 patients in the PMRT group and 363 patients in the no-PMRT group. After a median follow-up of 37 months, PMRT was independently associated with increased OS, but there was no significant differences in OS or BCSS between the groups stratified by the number of positive lymph nodes. Our study included 1172 patients after PSM with a median follow-up of 69.5 months. We speculate that the reason for inconsistent conclusions may be that they included fewer patients and had a shorter follow-up time. In addition, in their study, patients in the radiotherapy group had a heavier nodal burden, and the proportion of chemotherapy was higher than that in the non radiotherapy group. There was a significant difference between the two groups. There was no significant difference in baseline characteristics between the two groups in our study. This may also be the reason for our inconsistent conclusions. We have revised this section to provide a clearer explanation for the readers.

[1] Zhang L, Tang R, Deng JP, Zhang WW, Lin HX, Wu SG, et al. The effect of postmastectomy radiotherapy in node-positive triple-negative breast cancer. BMC Cancer. 2020; 20: 1146. doi: 10.1186/s12885-020-07639-x.

[2] Zhang L, Wang XX,  Lian JY, Song CG. Effect of postmastectomy radiotherapy on triple-negative breast cancer with T1-2 and 1-3 positive axillary lymph nodes: a population-based study using the SEER 18 database. Oncotarget. 2019;10(50): 5245-5252. doi: 10.18632/oncotarget.24703.

We tried our best to improve the manuscript. Thank you for the kind advice, and we hope that the correction will meet with approval.

Sincerely,

Lin-Yu Xia

---

## [Editor Report · Decision Letter 1]

12 Jun 2022

Effect of postmastectomy radiotherapy on T1-2N1M0 triple-negative breast cancer.

PONE-D-22-00975R1

Dear Dr. Xia,

We’re pleased to inform you that your manuscript has been judged scientifically suitable for publication and will be formally accepted for publication once it meets all outstanding technical requirements.

Kind regards,

Sudeep Gupta, M.D.

Academic Editor

PLOS ONE

Additional Editor Comments (optional):

The manuscript can be accepted for publication.
---

## [Editor Report · Acceptance letter]

16 Jun 2022

PONE-D-22-00975R1 

Effect of postmastectomy radiotherapy on T1-2N1M0 triple-negative breast cancer. 

Dear Dr. Xia:

I'm pleased to inform you that your manuscript has been deemed suitable for publication in PLOS ONE. Congratulations! Your manuscript is now with our production department. 

Kind regards, 

on behalf of

Dr. Sudeep Gupta 

Academic Editor

PLOS ONE